# Thermally trainable dual network hydrogels

Shanming Hu [1], Yuhuang Fang [1], Chen Liang[1], Matti Turunen[1], Olli Ikkala [1] ✉ & Hang Zhang [1] ✉

Inspired by biological systems, trainable responsive materials have received burgeoning research interests for future adaptive and intelligent material systems. However, the trainable materials to date typically cannot perform active work, and the training allows only one direction of functionality change. Here, we demonstrate thermally trainable hydrogel systems consisting of two thermoresponsive polymers, where the volumetric response of the system upon phase transitions enhances or decreases through a training process above certain threshold temperature. Positive or negative training of the thermally induced deformations can be achieved, depending on the network design. Importantly, softening, stiffening, or toughening of the hydrogel can be achieved by the training process. We demonstrate trainable hydrogel actuators capable of performing increased active work or implementing an initially impossible task. The reported dual network hydrogels provide a new training strategy that can be leveraged for bio-inspired soft systems such as adaptive artificial muscles or soft robotics.

Biological systems provide abundant inspirations for the design of synthetic materials with adaptive properties[1–5], such as the hypertrophy of muscles under repetitive (increasing) stress[6] and the strengthening of bones under high mechanical loads[7]. Such features allow adaptation to changing environmental stresses, thus improving the robustness and chance of survival in living organisms. This has inspired trainable artificial materials[8–11], which can adapt their mechanical properties to the experienced stimuli, in particular mechanical loads. Examples include the mechano-responsive double network (DN) hydrogels[8], muscle-like hydrogels based on the alignment of nanocrystalline domains[9], vibration-induced strengthening of a composite organo-gel[10], and cyclic stretching-induced reorganization of carbon nanotube yarns[11]. While these state-of-the-art systems exploit mechanical load as the stimulus to enhance the mechanical properties, they do not perform active work as in the case of real muscle. There exists a challenge to implement the training principle in responsive materials that are capable of active deformations triggered by, for instance, temperature or other types of stimuli. While there does not yet exist a unified definition, we broadly define trainable materials as materials capable of modifying their own properties and/ or stimuli-responses depending on the intensity of stimulus previously experienced, without the participation of a different stimulus or addition of chemicals. The modified properties can be, for instance,

mechanical, thermo-rheological, optical properties, or phase transitions. Though the trainable materials should possess a memory element of previous experiences, such memory can be distinguished from other types of memories developed in responsive materials[12–17], such as shape-memory[18], hysteresis-based memory[12,19], multiple phases[20,21], associative memory based on self-assembly[14], or swelling-induced memory[15]. The memories in these systems either require additional stimuli[14,15] or chemicals[17], or are only based on hysteresis in response to certain stimuli[12,20,21].

On the other hand, various mechanisms have been developed to post-modify or control the properties of responsive materials, such as temperature-triggered reversible hardening of a hydrogel[22], photo-triggered mechanical strengthening in synergistic covalent and supramolecular polymers[23], thermally induced self-healing and strengthening in an elastomer[24], post-modification of sample size and conductivity in organohydrogels by feeding matrix nutrients[25], thermal stiffening of a hydrogel by polymer-cluster interactions[26], toughening of a hydrogel by post-formation of metal-coordination complex[27], swelling-induced strengthening of a hydrogel by deformable nano-barriers[28], and toughening of a hydrogel by thermal treatment in air[29]. Another interesting artificial muscle system capable of performing active work has shown self-strengthening upon mechanical trainings[30]. However, the stimulus (heat) driving the deformation is different from the stimulus

[1]Department of Applied Physics, Aalto University, P.O. Box 15100, Espoo FI 02150, Finland. ✉e-mail: olli.ikkala@aalto.fi; hang.zhang@aalto.fi

(deformation) triggering the mechanical strengthening, and the thermal stimulus cannot enhance the mechanical properties alone. Despite the fascinating progresses in these systems with post-modification capability, there is generally a lack of training in the material's response by the stimulus driving the response.

Herein, we report a thermally trainable hydrogel system based on double network or interconnected interpenetrating network (IIN) design, where the size and thus thermal response of the system can be either increased or decreased through a training process above a certain temperature threshold. In this way, the temperature can control and train the response of the system in a positive or negative way. The kinetics and degree of training are investigated by varying the compositions and training conditions. The application potential is demonstrated by constructing hydrogel actuators that can be trained to perform increased active work or implementing a task impossible before training. The reported dual network hydrogels provide implications for bio-inspired soft systems such as thermally driven artificial muscles[31,32] or adaptive soft robotics[33].

## Results

### Design of thermally trainable hydrogels

Figure 1 illustrates the training processes to enhance or decrease the extent of thermoreversible swelling in the dual network hydrogels, unlike the fixed and reversible volume phase transitions in conventional responsive materials. Hereinafter we denote such processes as positive or negative training, suggested also for more general stimuli and trainable responses. Typically, the responses of materials depend on the magnitude of the stimuli (Fig. 1a, b). The response is reversible and fixed, i.e., it does not depend on the history of the stimuli. Famous examples include the phase transitions in hydrogels, which have been predicted by Dusek and Patterson in 1968[34,35] and first experimentally observed by Tanaka in 1978[36]. The crosslinked hydrogel network is non-ergodic[37], and hysteresis is commonly observed during phase transitions. In particular, poly(N-isopropylacrylamide) (PNIPAm) hydrogel undergoes reversible shrinking and swelling due to its Lower

Critical Solution Temperature (LCST) as shown in Fig. 1f[38]. Here we use LCST to denote the phase transition temperatures of the PNIPAm networks in our present system. In contrast, the trainable materials should be able to modify their response depending on the history of the stimuli, in particular when the stimulus exceeds a certain threshold. Here, we define the positive training as the process where a high-intensity stimulus will permanently enhance the subsequent response of the material, even when the intensity of the stimulus recovers, as shown in Fig. 1c. On the contrary, the negative training will decrease the response of the material upon training with a high-intensity stimulus (Fig. 1d). We will demonstrate in the following sections that both positive and negative training can be achieved in dual network hydrogels containing an agarose network and a PNIPAm network by the design of the network architecture.

To achieve positive training, we propose a double network (DN) hydrogel containing an agarose network and a chemically crosslinked PNIPAm network. Agarose is a physically crosslinked hydrogel with thermoreversible sol-gel transitions, where nanoscopic bundles of semiflexible fibrils are formed in the gel state compared to polymer coils in the sol state[39,40]. Typically, the sol-gel transition shows large hysteresis, where the melting temperature is dozens of degrees higher than the gelling temperature[41]. Agarose has been utilized to construct double network hydrogels with, e.g., polyacrylamide or PNIPAm, which showed thermoreversible mechanical properties[42,43] or enhanced whiteness upon phase transition[44]. In this study, the DN hydrogel is formed by the gelation of pristine agarose and subsequent radical polymerization of the PNIPAm in the presence of a crosslinker, poly(ethylene glycol) diacrylate (PEGDA, $M_n$ = 10000 Da). In this way, the thermo-response of the hydrogel depends on both networks. Upon heating above the LCST (~ 35 °C) of PNIPAm but below the melting point of agarose ($T_m$ ~ 63 °C, Fig. 1e), the DN hydrogel will undergo typical volume phase transition (Supplementary Fig. 1) around 35 °C. The volumetric change in this hydrogel is, however, strongly restricted by the relatively rigid agarose network[45], and the amplitude of the volume change is thus smaller than a single PNIPAm network with the

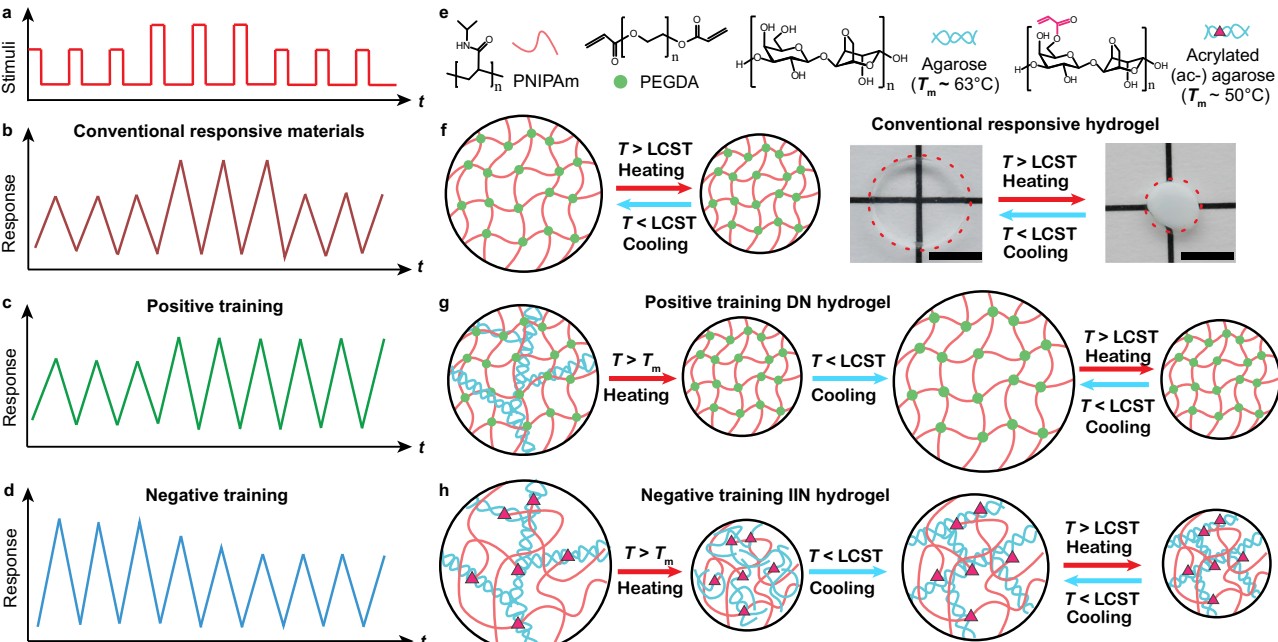

**Fig. 1 | Schematic illustrations of positive and negative training in dual network hydrogels. a** Stimuli of varying intensities. Responses of (**b**) conventional responsive materials, (**c**) positive training material, and (**d**) negative training material induced by the stimuli in (**a**). **e** Compositions of the trainable hydrogels. **f** A single network PNIPAm gel showing conventional thermoreversible volume changes corresponding to (**b**). Scale bar: 10 mm (black) for the photograph. **g** A double network (DN) agarose/PNIPAm hydrogel showing positive training corresponding to (**c**). **h** An interconnected interpenetrating network (IIN) ac-agarose/PNIPAm hydrogel showing negative training corresponding to (**d**).

same crosslinking density. In this temperature range, the agarose network remains in the gel state and the response of the DN hydrogel is thus reversible without changes. Upon a training process by heating up the hydrogel to a threshold temperature at 63 °C or above, the agarose network will melt into polymer coils and diffuse out of the PNIPAm network (Fig. 1g). After a few cycles of heating above $T_m$ and cooling to room temperature (RT, 25 °C), the agarose will be completely removed from the DN, as corroborated by FTIR measurement of pristine and trained hydrogels in Supplementary Fig. 2. Consequently, the PNIPAm network could swell and shrink to a much larger extent due to the removal of agarose network, even when heated well below the $T_m$. The thermal treatment above the threshold temperature thus can enhance the hydrogel's response in size, leading to positive training.

To implement negative training, we designed an interconnected interpenetrating network (IIN) hydrogel containing an acrylated agarose (ac-agarose) network and PNIPAm (Fig. 1h). The main difference of IIN to conventional DN or interpenetrating network hydrogel is that the two networks in the IIN are interconnected together by chemical bonds[46]. Hydroxyl groups on the agarose polymer chains are partially substituted by acrylate groups[47]. Therefore, the agarose can serve as the crosslinking points for the PNIPAm. The IIN hydrogel is formed by the gelation of ac-agarose and subsequent radical polymerization of NIPAm in the absence of any other crosslinkers. The two types of polymers are thus chemically interconnected, and the PNIPAm does not form an independent network, which is different from conventional double or interpenetrating networks[48,49]. Such IIN hydrogel shows a similar volume phase transition around 35 °C (Supplementary Fig. 3) as pure PNIPAm due to the collapsing of PNIPAm upon heating. In this hydrogel, heating above the threshold temperature, i.e., the melting point of ac-agarose ($T_m$ ~ 50 °C, Fig. 1e), results in both the shrinking of the PNIPAm and the melting of agarose fibrils (Fig. 1h). Due to chemical crosslinking, the agarose coils are trapped inside the hydrogel and cannot diffuse out of the hydrogel as confirmed by FTIR in Supplementary Fig. 2. Upon cooling to room temperature, which is below the gelling point of ac-agarose (~ 30 °C), the agarose re-gels inside the shrunken volume of the hydrogel, as the gelation of agarose (less than 1 min) is faster than the swelling of hydrogel (10–20 min, Supplementary Fig. 4) used in this system. This increases physical crosslinks per unit volume in the IIN hydrogel due to the higher effective concentration of agarose compared to the as-prepared state. Such reinforcement of the agarose network further restricts the swelling of the hydrogel, and the hydrogel's size decreases upon the thermal training, thus leading to a negative training process (Fig. 1h).

## Positive training hydrogels

Figure 2 demonstrates the positive training processes of DN hydrogel in response to various training conditions. Disk-shaped hydrogels were prepared with an as-prepared diameter ($d_0$) of 4 mm and thickness of 200 um (Fig. 2a). The diameters of the hydrogel disks at different temperatures are measured by an optical setup shown in Supplementary Fig. 5. The swelling ratio is defined as the ratio of equilibrium swollen/shrunken diameter ($d$) relative to the as-prepared diameter $d_0$. Each training cycle typically consists of a heating period of 5 min at high temperature (HT) followed by a cooling period of 30 min at room temperature (RT, 25 °C), unless otherwise mentioned. Both durations allow the hydrogels to reach swelling/shrinking equilibria, as shown by the kinetics measurement of the hydrogels in Supplementary Figs. 4 and 6, and Supplementary Table 1. The as-prepared hydrogels were always allowed to first reach swelling equilibrium at 25 °C, where the swelling ratio increased to between 1.46 and 1.85 depending on the composition of the hydrogel (Fig. 2). This is smaller than pure PNIPAm hydrogels (swelling ratio 2.2) prepared with the same crosslinker concentration as shown in Supplementary Fig. 7, demonstrating the restricting effect of the agarose network on the swelling of

the hydrogel. Figure 2a shows the typical response of hydrogels containing 2 wt% of agarose, 10 wt% PNIPAm, and 0.05 mol% PEGDA relative to PNIPAm. PEGDA is chosen as the crosslinker instead of conventional *N,N'*-methylenebisacrylamide (BIS) as it allows higher volume change upon LCST phase transition and a more significant training effect (Supplementary Fig. 8). The concentration of PEGDA is optimized to achieve the best combination of training effect and mechanical properties (Supplementary Figs. 9 and 10). Lower crosslinking density results in a higher training effect but mechanically rather weak hydrogels, while increasing the crosslinking density to 0.1 mol% results in a similar training effect compared to 0.05 mol% (Supplementary Fig. 9), but lower mechanical strength before training (Supplementary Fig. 10). Furthermore, 10 wt% of PNIPAm is chosen as it allows higher training effect compared to 15 and 20 wt%, as shown in Supplementary Fig. 11.

Upon initial heating at 40 °C, the DN hydrogel shrunk to 0.82 of the as-prepared size, but the response remained in a stable range without significant change during three heating-cooling cycles. Once heated to 63 °C, however, the hydrogel shrunk much more than at 40 °C to around 0.62, while the swelling ratio at 25 °C gradually increased to above 1.62 during three training cycles. When the HT returned to 40 °C after the training, the increase in response is almost fully preserved, demonstrating a permanent enhancement in the hydrogel's size at RT and response in Fig. 2a. The LCST response of the hydrogel between RT and 40 °C can be quantified by a response ratio $r_{LCST}$ in Eq. (1).

$$r_{LCST} = \frac{r_{s,RT}}{r_{s,40C}} \tag{1}$$

where $r_{s,RT}$ is the swelling ratio at RT (25 °C), and $r_{s,40C}$ is the swelling ratio at 40 °C. The $r_{LCST}$ in the pristine gel in Fig. 1a is 1.82, which corresponds to a volume increase of 600% at RT compared to the gel at 40 °C. In contrast, the $r_{LCST}$ in the trained gel is 2.81, corresponding to a volume increase of 2200% at RT compared to the gel at 40 °C. We further define the training ratio $r_{train}$ depending on the swelling ratio $r_s$ of the trained hydrogel (3 training cycles) to the $r_s$ of the pristine gel at RT as shown in Eq. (2).

$$r_{train} = \frac{r_{s,trained}}{r_{s,pristine}} - 1 \tag{2}$$

where the $r_{s,trained}$ and $r_{s,pristine}$ is the swelling ratio at RT of the trained hydrogel and pristine hydrogels, respectively. Therefore, the training ratio represents the change in the size of the hydrogel through the training process. Positive values represent an increase in the size of the gel, i.e., positive training, while negative values represent a decrease in the size of the gel, i.e., negative training. The hydrogel in Fig. 1a thus has a training ratio of 0.11.

Figure 2b summarizes the effect of different agarose concentrations on the training and response ratios during the training process at 63 °C, while the swelling ratio changes during the training process are shown in Supplementary Fig. 12. With increasing agarose concentrations from 1 to 3 wt%, the initial equilibrium swelling ratio at RT decreases from 1.77 to 1.53 (Supplementary Fig. 12) due to the more dense and rigid agarose networks formed at higher concentrations. Through repeated thermal trainings at 63 °C, the hydrogels undergo a gradual increase in the size and response ratio, which equilibrates after 3 cycles. The training ratios of the 1, 2, and 3 wt% agarose are 0.19, 0.13, and 0.11, respectively. On the other hand, the response ratios also show an increase after the training process as shown in Fig. 2b.

Figure 2c shows the effect of training temperature on the training and response ratios in DN hydrogels containing 2 wt% agarose, while the swelling ratio change during the training process is shown in Fig. 2d. If 40 °C is applied as the HT during training, the response of the

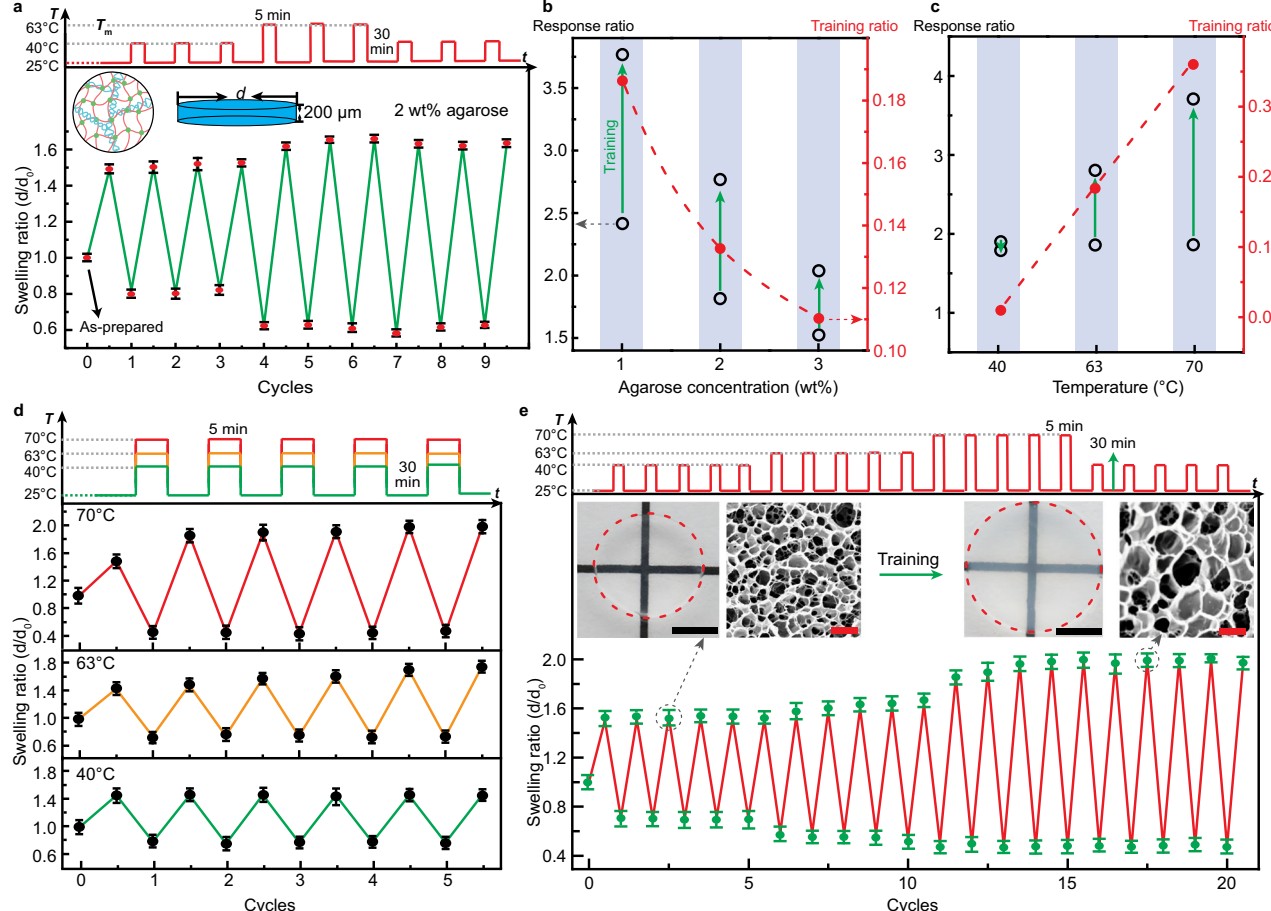

**Fig. 2 | Positive training of double network agarose/PNIPAm hydrogel. a** The swelling ratio change in response to heating-cooling cycles, showing the increase in response by training above the threshold temperature. **b** Effect of agarose concentrations on the training and response ratios. **c** Effect of training temperature on the training and response ratios. Green arrows indicate a change in response ratio after training. Red dashed lines are to guide the eye. **d** Swelling ratio change of hydrogels undergoing training at different temperatures. **e** Stepwise training of the hydrogel. Insets show photographs and SEM images of the hydrogel before and after training. Scale bar: 10 mm (black) for the photograph and 1 μm (red) for SEM images. Error bars represent standard deviations of measurement from four samples. Training cycle consists of 5 min at HT and 30 min at RT.

hydrogel almost remained constant, and the training ratio is practically zero. At 63 °C, one can observe a gradual increase of the response, which reaches a steady state after 3–4 cycles (Fig. 2d). If a temperature much higher than the melting point is used, i.e., 70 °C, the training process takes around 2 cycles to complete (Fig. 2d). The training ratio increases with the training temperature, reaching 0.36 at 70 °C, compared to 0.13 at 63 °C (Fig. 2c). On the other hand, the response ratio change is also higher at 70 °C, increasing from 1.9 to 3.7. Overall, higher training temperature leads to higher training ratios and an increase in the response ratio.

The different effects of temperature can be combined to perform stepwise training of the hydrogel, as shown in Fig. 2e. The hydrogel containing 2 wt% agarose underwent five cycles of thermal treatment at 40, 63, and 70 °C, respectively. The stepwise increase of response can be clearly observed at each temperature, with 70 °C producing the highest response and size increase. The response ratio increased from 1.9 to 3.6 after training, while the training ratio reached 0.35 by stepwise training. The training ratio is similar to the direct training (0.36) at 70 °C in Fig. 2c, d. The insets in Fig. 2e show the scanning electron microscope (SEM) images of the DN hydrogel before and after training at 70 °C, which clearly demonstrates an increase in the pore size in the trained hydrogel. This corroborates the effect of the agarose network in restricting the swelling of the PNIPAm network, where the trained hydrogel can swell to a much larger degree upon removal of the agarose. Full-scale SEM images of the DN hydrogel before and after

training can be found in Supplementary Fig. 13. The kinetics of the training process under different conditions have been studied by fitting the swollen size during training, and the results are shown in Supplementary Fig. 14, and Supplementary Table 2. The time constant for different training processes is between 0.7 and 2.0 cycles, where higher temperature and higher agarose concentrations lead to a faster training process (smaller time constant).

**Negative training hydrogels**

We next show in Fig. 3 the negative training process in the IIN hydrogel containing acrylated (ac-)agarose and PNIPAm. Due to the acrylation, the melting point of the agarose drops to around 50 °C, which is used as the training temperature shown in Fig. 3a. The degree of acrylation (DA) is defined as the feed ratio (in percentage) of acryloyl chloride to the total number of hydroxymethyl groups during the modification process, see Supplementary Fig. 15. DA 15 indicates that 15% of acryloyl chloride relative to the available hydroxymethyl groups was added during the modification process. The real degree of substitution has been determined using NMR as shown in Supplementary Fig. 15 and Supplementary Table 3. As an example, DA 15 results in a degree of substitution of 1.1%, while DA 50 leads to a degree of substitution of 2.8%. However, the DA (feed ratio) is used throughout this manuscript for consistency as the degree of substitution cannot be determined for very low feed ratios, such as 1% DA. For the hydrogel containing 2 wt% of ac-agarose with a DA of 15%, the thermal training below the melting

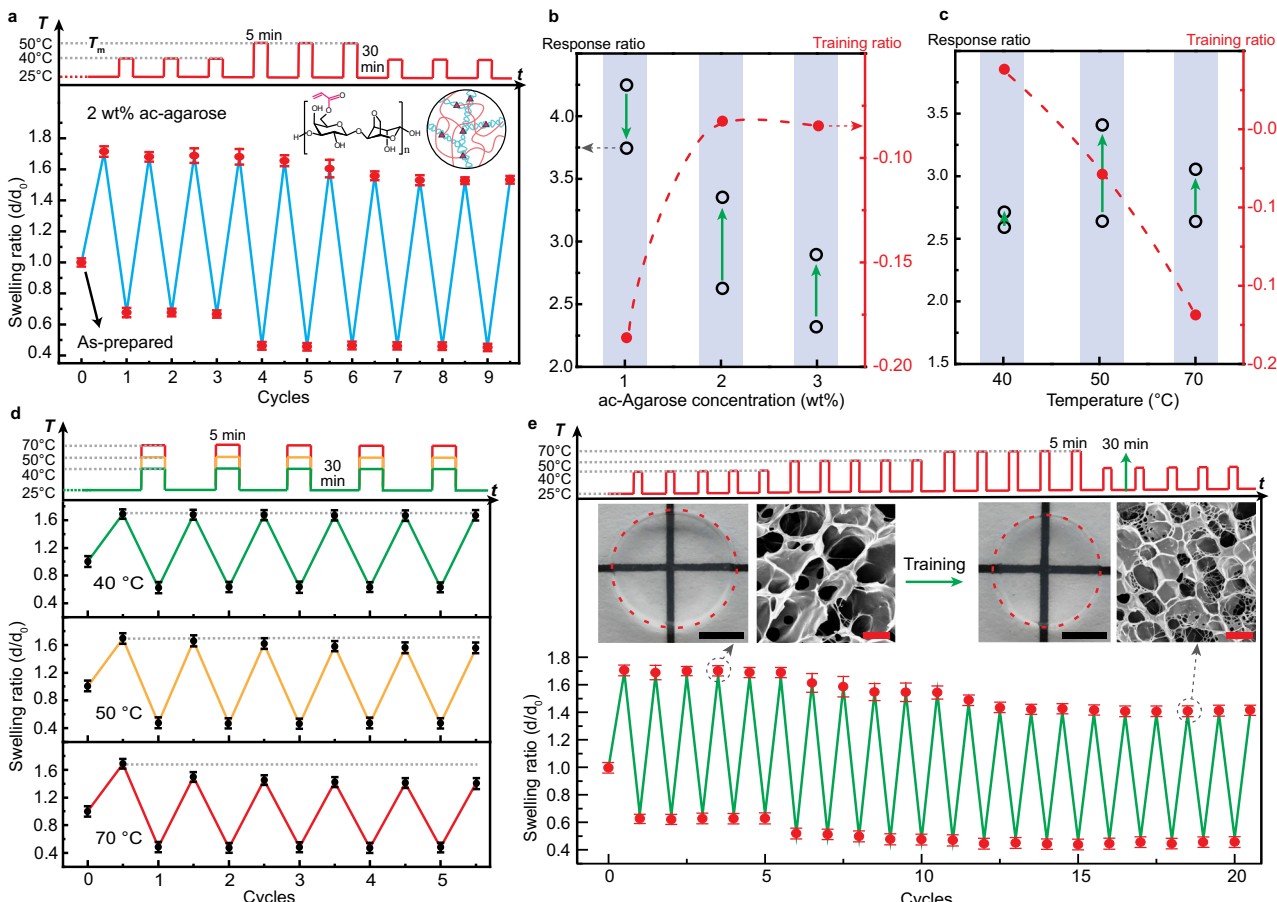

**Fig. 3 | Negative training of interconnected interpenetrating network ac-agarose/PNIPAm hydrogel. a** The swelling ratio change in response to heating-cooling cycles, showing a decrease in response by training above the threshold temperature. **b** Effect of ac-agarose concentrations on the training and response ratios. **c** Effect of training temperature on the training and response ratios. Green arrows indicate change of response ratio after training. Red dashed lines are to guide the eye. **d** Change of swelling ratio during training at different temperatures. **e** Stepwise training of the hydrogel. Scale bar: 10 mm (black) for photograph and 1 μm (red) for SEM images. Error bars represent the standard deviations of four measurements. Unless otherwise mentioned, the hydrogels contain 2 wt% ac-agarose (DA 15). Training cycle consists of 5 min at HT and 30 min at RT.

point, i.e., 40 °C, results in a minor decrease of size in the first cycle, where the swelling ratio at 25 °C changed from 1.79 to 1.71 while the swelling ratio at 40 °C remained constant at 0.63. The decrease is possibly due to the removal of unbound polymers and the physical contact of agarose chains during the shrinking of PNIPAm which increases physical crosslinks. When the training temperature further increased to 50 °C, there is a significant drop in both the swollen and shrunken size to 1.53 and 0.45, respectively. Such decrease is preserved when the temperature returns to 40 °C, and the reduced equilibrium swelling ratio at 25 °C remains stable for more than 30 h (Supplementary Fig. 16). The training ratio is thus −0.08, while the response ratio actually increased from 2.5 to 3.4 due to the decrease of the shrunken size. The effect of DA in the range between 1 and 50% on the training results has been summarized in Supplementary Fig. 17, where the 15% shows the most significant decrease of size under training at 50 °C.

Figure 3b demonstrates the effect of agarose concentration on the training and response ratios, while the swelling ratio change during the training process at 50 °C of these IIN hydrogels can be found in Supplementary Fig. 18. The concentration range between 1 and 3 wt% ac-agarose (DA 15) has been studied. Below 1 wt% ac-agarose, the hydrogel is too fragile and cannot withstand cyclic temperature changes, while the size changes of hydrogels are negligible during training for concentrations above 3 wt%. With increasing concentrations of ac-agarose, the initial swelling ratio at 25 °C decreases (Supplementary Fig. 18) due

to higher crosslinking density and rigidity of the agarose scaffold. The shrunken size at 50 °C is roughly the same for all concentrations at a swelling ratio of around 0.45. This tendency is similar to the positive training DN hydrogels, demonstrating the stronger effect of agarose in restricting the swelling of PNIPAm compared to shrinking. The training effect is most pronounced at 1 wt% ac-agarose, where the swelling ratio at 25 °C decreased from 2.01 to 1.63 after 3 training cycles, resulting in a training ratio of −0.19. The response ratio decreased from 4.2 to 3.7 after training. At 2 and 3 wt% of ac-agarose, the training ratios are −0.08 and −0.09, respectively, and the response ratios increased after training. The shrunken size of the hydrogels at 50 °C remained almost constant throughout the training process.

Figure 3c presents the effect of temperature on the training ratio and response ratio of hydrogels containing 2 wt% ac-agarose (DA 15), while the swelling ratio change during training can be found in Fig. 3d. It can be observed that 70 °C produces the most significant decrease of size during the training process, as the swelling ratio at 25 °C decreased from 1.67 to 1.39 after 4 cycles of training (Fig. 3d), resulting in a training ratio of −0.17. The shrunken size at 70 °C remained almost constant at 0.45, smaller than that at 40 °C (0.63). However, due to the decrease of the swollen volume at 70 °C, the response ratio change is also smaller than that at 50 °C (Fig. 3c). The training ratios for 70 °C and 50 °C are thus −0.17 and −0.08, respectively. At 40 °C below the melting point, no significant change in the size was observed for the five cycles of training, and the training ratio was only −0.01.

The stepwise negative training of the IIN hydrogel is demonstrated in Fig. 3e, where 40, 50 and 70 °C trainings (5 cycles each) have been applied consecutively to the hydrogel. As the training temperature increased from 40 to 50 °C, the shrunken size of the hydrogel above LCST decreased from 0.66 to 0.48, while the swollen volume at 25 °C underwent a slight reduction from 1.58 to 1.49. When the training temperature further increased to 70 °C, the swollen size of the hydrogel further decreased to 1.39, and the reduced size is preserved in the hydrogel when the training temperature returned to 40 °C. The final training ratio after the stepwise training was −0.16, almost the same as in single training (−0.17) at 70 °C in Fig. 3c. The insets in Fig. 3e show the SEM images of the IIN hydrogel before and after training at 70 °C, which demonstrates a decrease in the pore size in the trained hydrogel. This reflects the more densely crosslinked agarose network in the trained hydrogel, which further restricts the swelling of the PNIPAm network and thus reduces the pore size. Full-scale SEM images of the IIN hydrogel before and after training can be found in Supplementary Fig. 13. Compared to the agarose/PNIPAm DN hydrogel, the pores in the pristine ac-agarose/PNIPAm IIN hydrogel are larger due to the lack of PEGDA crosslinker, and the PNIPAm network in the IIN hydrogel is only crosslinked by the acrylate groups in the ac-agarose. The kinetics of negative training under different conditions have been studied by fitting the swollen size during training with exponential decay function, and the results are shown in Supplementary Fig. 19 and Supplementary Table 4. The time constant for different training processes is between 0.8 and 1.6 cycles.

## Training of mechanical properties

In addition to the training of volumetric response, the mechanical properties of the hydrogels are also strongly influenced by the training

process, as exemplified in Fig. 4. For the positive training hydrogel shown in Fig. 4a, the DN containing 2 wt% agarose softens after the training at 70 °C due to the removal of the agarose network. Besides, the hydrogel also becomes more fragile due to the significantly increased swelling, where the swollen volume increased by 2100% as shown in Fig. 2a. The modulus measured in the strain range between 0 and 0.5 decreased from 105 kPa to 4 kPa, and the ultimate tensile strength decreased from 63 kPa to 7 kPa. Despite the softening of the hydrogel, the significant increase in volumetric response upon phase transition can be utilized to perform active work. One example is shown in Fig. 4d, where repeated lifting of a 5 g weight was performed on a cylindrical hydrogel with an as-prepared diameter of 5 mm. Upon loading of the weight, the length of the swollen hydrogel was stretched from 5.6 cm to 7.3 cm. At 40 °C, the weight underwent small upwards displacement for around 1 cm as the hydrogel shrinks, which corresponds to a work of 0.35 mJ. The hydrogel was allowed in each training cycle to swell for 3 h at RT to reach equilibrium due to the increased size. After two cycles of training at 70 °C, the softening of hydrogel leads to an increase of the elongation to 11 cm, while the shrunken length increased slightly to 7.6 cm. In this way, the positive training hydrogel can perform more work during the volume phase transition at same temperature (40 °C), which increased from 0.35 mJ to 1.7 mJ after three cycles of training.

On the other hand, the negative training IIN hydrogels do not experience as large size changes during the training process as positive training hydrogels. However, the mechanical properties are strongly affected by the training process due to the reformation of physical crosslinks in the agarose network. For the hydrogel containing ac-agarose (DA 15), an increase of 80% in ultimate tensile strength (UTS) after the training is observed, though the modulus remained roughly

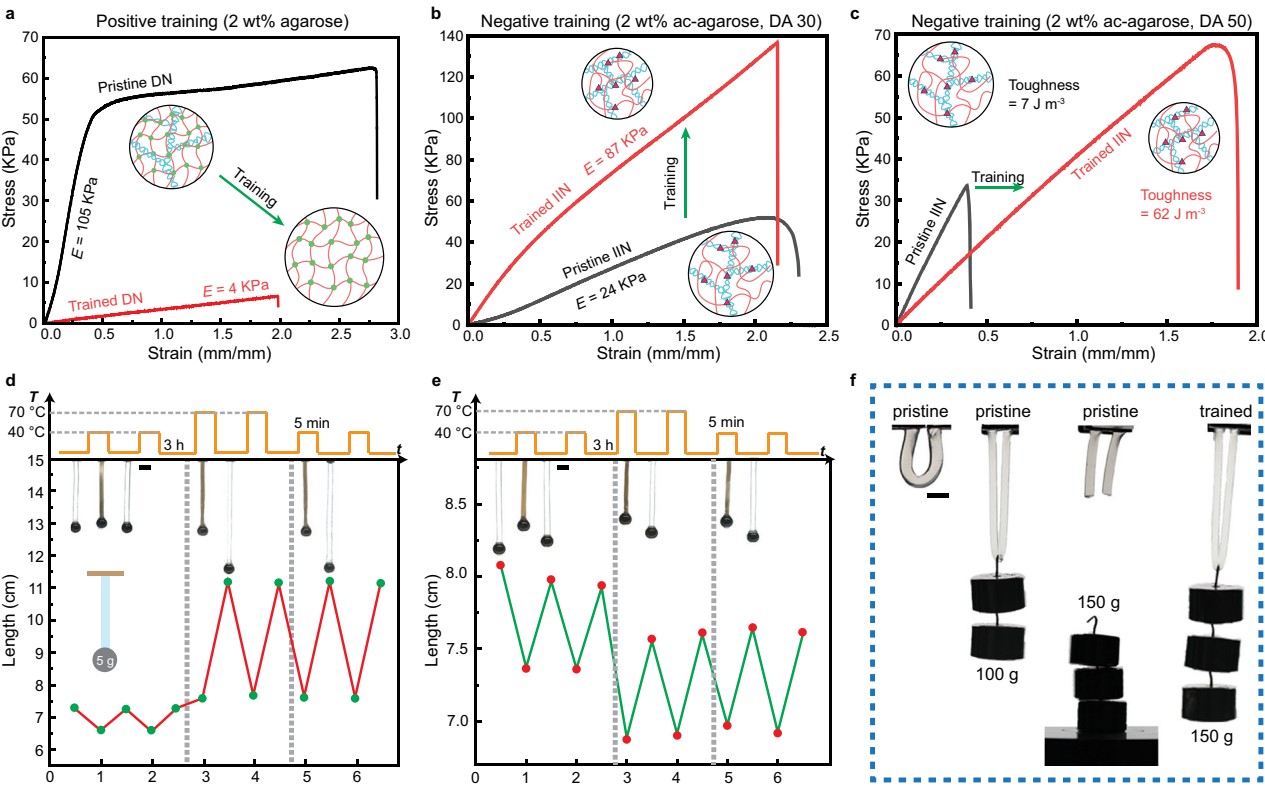

**Fig. 4 | Training of mechanical properties. a** Tensile tests of positive training hydrogel containing 2 wt% agarose, showing softening after training. **b** Tensile tests of negative training hydrogel containing 2 wt% of ac-agarose (DA 30), showing enhancement of Young's modulus (stiffening) after training. **c** Tensile tests of negative training hydrogel containing 2 wt% of ac-agarose (DA 50), showing

increase in toughness (toughening) after training. **d** Active weightlifting of a 5-gram metal sphere in a positive training hydrogel corresponding to (**a**). **e** Active weightlifting of a 5-gram metal sphere in a negative training hydrogel corresponding to (**b**). **f** Passive weightlifting of negative training hydrogel corresponding to (**c**). All trainings were conducted at 70 °C for 3 cycles. Scale bars: 10 mm.

constant, see Supplementary Fig. 20. Figure 4b shows the IIN hydrogel containing 2 wt% ac-agarose (DA 30) before and after training, where the modulus and UTS increased significantly while the strain at rupture remains constant. The modulus increased from 24 to 87 kPa after three cycles of training at 70 °C, while the UTS reached 137 kPa compared to 52 kPa before training. The stiffening effect of this hydrogel is shown also in Fig. 4e, where the length of the cylindrical hydrogel at 25 °C decreased from 7.9 to 7.6 cm after training. The work performed during each phase transition cycle is not much affected by the decrease of the hydrogel's volume, since the shrunken size of the hydrogel also decreased.

Intriguingly, the toughness of the IIN hydrogel can also be dramatically improved by the training process, as shown in Fig. 4c. The hydrogel contained 2 wt% ac-agarose with a DA of 50. After the training process, the modulus decreased from 88 to 43 kPa, while the toughness increased from 7 J m$^{-3}$ to 62 J m$^{-3}$. The UTS is doubled in the trained hydrogel compared to the pristine hydrogel. Note that the swollen size of the hydrogel remained almost constant for such hydrogels (Supplementary Fig. 17). This is in contrast to reported mechanical training systems,[8,9] where significant elongations are typically observed during the training process. The toughening effect is shown in Fig. 4f, where the pristine hydrogel can only hold 100 g of weight. After the thermal training, the hydrogel withstands 150 g of weight without breaking. It should be noted that the volume change of PNIPAm during heating plays an important role in enhancing the mechanical properties, as discussed in the previous section. This is corroborated by the control experiment of an IIN hydrogel prepared using acrylamide instead of NIPAm, which showed only a minor change in mechanical properties by the same training process (Supplementary Fig. 21).

## Thermally trainable actuators

The application potential of the trainable hydrogels as soft actuators are demonstrated in Fig. 5. To convert isotropic volume change into deformations, bilayer hydrogels are synthesized, which contained a layer of positive training DN hydrogel (2 wt% agaroses) and a layer of negative training IIN hydrogel (2 wt% ac-agarose, DA 15). The as-prepared thickness of each layer is 0.5 mm, which is linked together by the interpenetration of networks at the interface during polymerization. The negative training hydrogel contained Rhodamine dyes for better visibility. Detailed preparation methods can be found in the experimental section. Due to different swelling ratios and mechanical properties of the two layers, the as-prepared bilayer underwent minor bending towards the positive training hydrogel side, corresponding to negative bending angles as shown in Fig. 5a. Once heated above the LCST, the negative training hydrogel shrunk more than the positive training hydrogel (c.f. Figs. 2 and 3), which caused a reverse bending with a positive bending angle. The bending angle is defined as the angle between the direction of the free end and the vertical direction shown in Fig. 5a. The photographs of the bilayer actuator and the corresponding bending angle during a single training step are shown in Fig. 5b, c. Note that the x-axis is not uniform in order to show the deformations during heating and cooling periods. The actuator underwent much higher bending at 70 °C than at 40 °C, where the bending angle increased from 63 to 171 degrees. Such an enhanced bending deformation is preserved in the soft actuator after the training process, which shows an increase in bending angle to 154 degrees at 40 °C. On the other hand, the bending angle at RT also increased from −20 to 30 degrees due to the increased swelling ratio of the positive training hydrogel.

Based on the bilayer concept, we further constructed a four-arm gripper made of the same hydrogel bilayers, where the negative training hydrogel faces the objects to be grabbed. The pristine gripper at 25 °C shows a reverse bending as shown in Fig. 5d. When heated to 40 °C, the gripper cannot grab a metal sphere (0.5 g, diameter 5 mm) as the closure of the arms is not sufficient. With a single training step of heating to 70 °C for 5 min, the gripper's performance is notably enhanced. At 40 °C after training, the bending of each arm is sufficient to enclose the metal sphere, which could be lifted up and transported

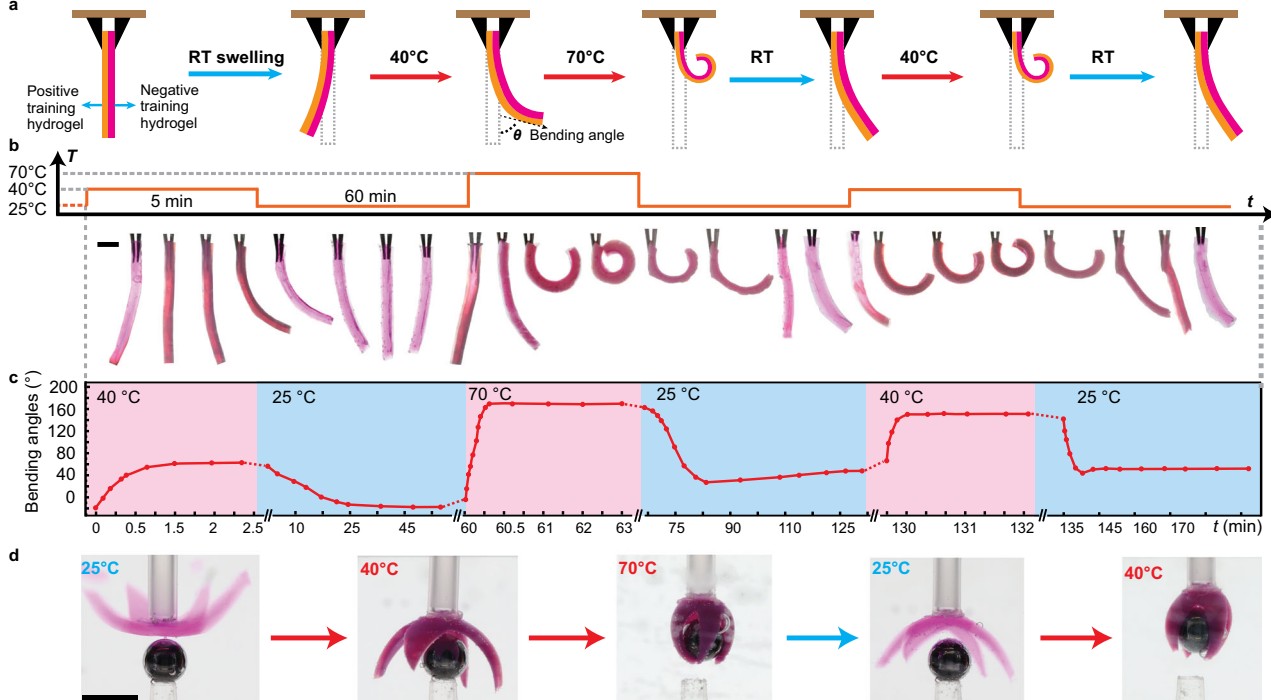

**Fig. 5 | Trainable actuators. a** Illustration of a bilayer actuator consisting of positive and negative training hydrogels. The as-prepared thickness of each layer: 0.5 mm. **b** Photos of the actuator's deformation during training. Scale bar: 2 mm. **c** Change of bending angle during training. Note the x-axis is not uniform. **d** Training of a gripper consisting of four bilayer fingers. The red dye was added to the hydrogels for better visibility. Scale bar: 10 mm. Positive training hydrogels contained 2 wt% agarose. Negative training hydrogels contained 2 wt% ac-agarose (DA 15).

as shown in Fig. 5d and Supplementary Movie 1. A similar four-arm bilayer actuator is shown in Supplementary Fig. 22, demonstrating the enhanced closure of the arms after training at 70 °C. The above demonstrations show the potential of the thermally trainable hydrogels to be used as soft and adaptive actuators, where the performance can be enhanced by the past experience.

## Discussion

We have designed and synthesized thermally trainable hydrogels using DN or IIN networks consisting of PNIPAm and agarose or acrylated ac-agarose. Training towards increased (positive) or decreased (negative) swollen size can be achieved through a series of heating and cooling cycles above certain threshold temperature. At temperatures below the melting points of the agarose or ac-agarose, the hydrogels show reversible and stable size changes to heating similar to conventional single network PNIPAm. Once the temperature exceeds the melting point (e.g., 50 or 63 °C), melting or reconstruction of the agarose networks will cause either weakening of strengthening of the hydrogel, leading to permanent modification of the swelling equilibrium. After training, the enhanced or reduced responses are preserved upon stimulus below the threshold temperature. The training ratio and response ratio can be controlled by the composition of the hydrogel and the training temperature. In addition to the volumetric response, the mechanical properties can also be notably modified by the thermal training process. The positive training hydrogels show significant softening, whereas the negative training hydrogels can undergo stiffening or toughening depending on the degree of acrylation. Application potentials of the trainable hydrogels as active soft actuators are demonstrated by the capability of performing more mechanical work or undertaking an initially impossible task after training. The reported DN and IIN hydrogels provide a new strategy for designing trainable material systems that can be leveraged for bio-inspired soft systems such as thermally driven artificial muscles or adaptive soft robotics.

## Methods

### Materials

$N$-isopropylacrylamide (NIPAm, 99%) monomer was purchased from Fisher Scientific and recrystallized for three times in n-hexane (≥97%, Sigma-Aldrich). Acrylamide (AAm, 99%), agarose (low gelling temperature, A4018), poly(ethylene glycol) diacrylate (PEGDA, $M_n = 10\,000$), $N$, $N'$-Methylenebisacrylamide (BIS, ≥99.5%), 2-hydroxy-4'-(2-hydroxyethoxy)−2-methylpropiophenone (Irgacure 2959, photoinitiator, 98%), Rhodamine B, $N,N$-dimethylacetamide (DMAc, 99%), acryloyl chloride (97%), dimethyl sulfoxide-d6 (99.9 atom% D), and acetone (99.5%) were purchased from Sigma-Aldrich. Deionized water (18.2 MΩ, DirectQ 3 UV, Millipore) was used throughout the experiments.

### Modification of agarose

1.0 g of agarose was dissolved in 25 mL of DMAc at 100 °C. The agarose solution was then cooled to 0 °C in an ice-water bath, and a suitable amount (25, 38, 75, and 125 μL) of acryloyl chloride diluted in 1 mL DMAc was added slowly under stirring. The mixture was continuously stirred at 0 °C for 1 h and then at room temperature for 4 h. The product was precipitated and washed thoroughly by acetone, and then dried in vacuum.

### Preparation of dual network hydrogels

The positive and negative training hydrogels are prepared with the same protocol. The stock solutions of 2.5 wt% agaroses or ac-agarose were first prepared by dispersing 25 mg of agarose or ac-agarose in 975 μL deionized water at room temperature, which was then heated by a heat gun and vortexed until a clear solution was obtained. Subsequently, 100 mg of NIPAm monomer, 1.98 mg of photoinitiator Irgacure 2959, and optionally 4.46 mg crosslinker PEGDA for positive

training hydrogels were dissolved in 100 μL deionized water and 800 μL of the agaroses stock solution at around 35 °C by vortexing. The final solution is thus composed of 10 wt% NIPAm, 2 wt% agarose or ac-agarose, and for positive training hydrogel also 0.05 mol% of PEGDA relative to NIPAm. The hydrogels with other concentrations of agarose were prepared following the same protocol, except that a 3.5 wt% stock solution was prepared for a final agarose concentration of 3% in the hydrogels. The solution was bubbled with nitrogen to remove oxygen. The degassed solution was injected into a mold between two glass substrates separated by a spacer of 0.2 mm made of Parafilm (Parafilm M, Bemis). The mold was kept in a glass bottle degassed with nitrogen for 30 min and then stored in a fridge at 4 °C for 30 min for the gelation of agarose or ac-agarose. The polymerization of the hydrogel was finally carried out in a UV chamber (8 × 14 W lamps, 350 nm, Rayonet, USA) for one hour. The control sample containing AAm instead of NIPAm was prepared in the same way.

### Swelling/shrinking test of hydrogels

After polymerization, the hydrogel films were cut by a puncher (diameter 4 mm, the as-prepared size) to obtain disk-shaped hydrogels. The hydrogels were immersed in a large amount of deionized water overnight to remove unreacted monomers and to reach the equilibrium swollen state. The hydrogel disks were loaded in a rectangular glass tube (inner dimension: 1 mm × 10 mm × 50 mm, CM Scientific) filled with water, which was sealed by parafilm. To study the response of the hydrogels under thermal treatment, the glass tubes loaded with trainable hydrogels were placed on a thermoelectric temperature-controlled stage (TE Technology, TC-720). The temperature of the glass tube was calibrated by an infrared thermal camera (FLIR C3). Typically, the thermal training consists of 5 min equilibration at high temperatures (e.g., 50, 63 or 70 °C) and 30 min of equilibration at room temperature. The sizes of the hydrogels were recorded by a digital camera (Canon 60D). Finally, the size of the hydrogel disks was analyzed by the *Analyze Particles* function in Image J (v. 1.53q). Each data point was averaged from 4 pieces of samples, and the error bars represented the standard deviations.

### FTIR spectroscopy

The samples were frozen in liquid nitrogen and dried in a lyophilizer (Labconco Corporation) under 0.03 mbar for three days. The lyophilized samples were characterized by FTIR spectroscopy (ALPHA II, Bruker) with a Platinum-ATR-sampling module to obtain the spectra.

### Mechanical tests

The tensile tests of hydrogels were performed on an Instron 5567 universal testing machine at a strain rate of 100% min⁻¹.

### NMR characterization

$^1$H-NMR spectra were recorded on a Bruker NMR Spectrometer Avance III 400 (400 MHz), using dimethyl sulfoxide-d6 as the solvent.

### Scanning electron microscopy (SEM) characterizations

The cross-section of the hydrogel was obtained by cutting the hydrogel in liquid nitrogen, which was then freeze-dried for 24 h. The samples were coated with 4 nm Au-Pd, and the SEM images were recorded by Sigma VP (Zeiss, Germany).

### Fabrication of trainable bilayer hydrogel actuators

The as-prepared bilayer hydrogel consists of 0.5 mm of positive training hydrogel and 0.5 mm of negative training hydrogel, bound together by physical interpenetration at the interface. Firstly, a mold made of two glass slides was prepared using Parafilm as the spacer with a thickness of 0.5 mm. The precursor solution of positive training hydrogel was added, which contained an additional 0.3 wt% of Rhodamine B relative to the NIPAM for visibility. After the gelation of

agarose in a fridge for 30 min, the old spacer was removed, and a new spacer of 1 mm thickness was added, which forms the new space of 0.5 mm for the second layer. The precursor solution of negative training hydrogel was then injected and gelled again in a fridge for 30 min. Finally, the hydrogel bilayer was polymerized in a UV chamber (8 × 14 W lamps, 350 nm, Rayonet, USA) for one hour under nitrogen protection. The synthesized hydrogel films were finally cut into 1 × 5 × 20 mm rectangular shapes and immersed in abundant deionized water for one day to remove impurities and reach swelling equilibration.

## Data availability

The authors declare that all data supporting the findings of this study are available within the paper and its Supplementary Information files or available from the corresponding authors upon request. Source data are provided with this paper.

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

## Acknowledgements
We thank the provision of facilities and technical support by Aalto University at OtaNano - Nanomicroscopy Center (Aalto-NMC). We acknowledge fundings from Academy of Finland (Postdoctoral Researcher No. 331015 to H.Z., and Center of Excellence in Life-Inspired Hybrid Materials - LIBER No. 346108 to O.I.), China Scholarship Council (No. 202207960015 to Y.F.) and the European Research Council (Advanced Grant DRIVEN No. 742829 to O.I.).

## Author contributions
H.Z. conceived the idea and designed the experiments. H.Z. and O.I. supervised the project. S.H. carried out the experiments with assistance from Y.F., M.T., and C.L. H.Z. and S.H. analysed the data. H.Z., S.H., and O.I. wrote the manuscript with input from others.

## Competing interests
The authors declare no competing interests.
