## [Peer Review File · Nature Communications]

Thermally Trainable Dual Network HydrogelsReviewers' Comments:

Reviewer #1:

Remarks to the Author:

Recommendation: Publish in Nature Comm after minor revisions (almost as is)

The article by Hu et al. reports the design of thermally trainable hydrogel systems based on a double or mixed network design. The volumetric response of the system can either increase in volume (positive training) or decrease in volume decrease (negative training) through a training process induced by temperatures above either the cloud point of the PNIPAm network or above the melting point of the contained agarose network.

Positive training is achieved by heating the dual networks, melting the agarose, and diffusing the agarose out of the networks, which allows the PNIPAm network to swell and de-swell more freely and results in a larger swollen state.

For negative training, a mixed network containing PNIPAm and acrylated agarose is used. Due to chemical cross-linking, the agarose coils become entrapped in the hydrogel upon heating and contraction. Upon cooling, the agarose gels faster than the PNIPAm reswells, so it is trapped in a swelling-restricted state, resulting in an overall shrunken state.

Using tensile tests, the authors demonstrated that mechanical properties can be modified by temperature-induced training. Here, positive training leads to softer hydrogels and negative training leads to tougher and stronger hydrogels. For demonstration, the authors used a gripper consisting of a bilayer of both trainable systems, and thus making a gripper that must first needs to be conditioned at high temperatures to be fully functional at lower temperatures.

Overall, this is an excellent paper that clearly presents its content. The data provided data is nearly flawless. Although it may be bold to simply to wash agarose out of a double network and call this positive training, it is applied in a clever way. The bilayer actuator that needs to be conditioned to work at lower temperatures is impressive.

Minor points:

1. You state that you are using 0.05 mol% PEGDA in relation to NIPAm. Is this an optimized ratio? Because the CL density impacts the swelling of pNIPAm and thus the training ratio of the positive training hydrogels. If you have data on this, it should be provided in the supporting information.
2. With respect to Fig. S6: One might wonder why the initial swelling ratio at rt is not the same before heating to 40 and 70 °C.

Great Paper! Congratulations

Reviewer #2:

Remarks to the Author:

In this manuscript, the authors demonstrate a thermally trainable hydrogel system based on Agarose/PNIPAm DN gel. Either positive or negative training can be achieved, and the mechanical properties after training lead to soften, stiffen or toughen the gels. The soft hydrogel actuators were also prepared based on the bilayer gel of positive and negative gels. The work is interesting, and the design is ingenious. I'd like to recommend to accept this work by Nature Comm after the authors addressing well the following issues:

1. Since the network of gels are mainly depended on double network hydrogels, the title using "Dual Network Hydrogels" isn't suitable.
2. The training concept used in this work is much different from others because the final hydrogels were compared not with the original DN gel but with the gel after removal of agarose. If compared to the original DN gel, all the mechanical properties became very poor. Therefore, I suspect whether the concept "training" is correctly used or not?
3. In Figure 1h, the authors introduced the ac-agarose in the hydrogel to achieve negative training. But it easily makes reader confuse. It seems that additional ac-agarose was added based on Figure

1e.

4. The negative training is achieved by the rapid gelation of agarose while the slow chain dynamic of PNIPAm as the temperature $>$ LCST. The agarose first network resists the chain extension or recovery. However, if the time is long enough, can the size of gel can be recovered to the original one?

5. In Figure 3e, the SEM before training is for ac-Agarose/PNIPAm DN gel? If so, compared to Figure 2e with Agarose/PNIPAM DN gel, why ac-Agarose/PNIPAm DN gel with chemical crosslinking show much looser network structure at the same concentration of agarose?

Reviewer #3:

Remarks to the Author:

(1) The abstract starts with the following sentence: "Inspired by biological systems, mechanically trainable materials have received burgeoning research interests as they harbor great potential for future adaptive and intelligent material systems."

The literature is full of clichés like this. Is it really necessary to start the abstract with this cliché? The reviewer has similar feelings about the use of the word "training". The reviewer realizes that the (ab)use of renaming existing concepts is becoming an unfortunate habit in the literature. But there is no need to rename thermo-rheological (or physical) history that the material experienced to "training". The reviewer's strong advise is to please reconsider the language and use established, classical terminology whenever possible, instead of the fancy looking (and sounding) language that does not help understanding. This comment is just a friendly advise.

(2) However, the abstract is full of such blah-blah, and does not really serve the function of a scientific abstract. An abstract should be "...a summary of the contents of a book, article, or speech" (Oxford dictionary). This "abstract" is thus not an abstract, as there is virtually nothing about the summary of the article. Conclusion: please rewrite and make it a real abstract (what/why/how... you studied and what were the main findings/conclusions).

(3) Regarding the introduction it is very disappointing to see that the authors did not consider the pioneering work of Tanaka et al. (Annaka, M., Tanaka, T. Multiple phases of polymer gels (1992) *Nature*, 355 (6359), pp. 430-432, which is Tanaka's first article in his pioneering oeuvre. The reviewer would like to point out another paper, which describes the effects of physical history (without calling "training"): Annaka, M., Tokita, M., Tanaka, T., Tanaka, S., Nakahira, T. The gel that memorizes phases (2000) *Journal of Chemical Physics*, 112 (1), pp. 471-477.

(4) Similarly disappointing that the authors did not include in their introduction and concepts, the work by Dusek et al. about the transitions in swollen polymer networks: Dusek K., Patterson D. (1968) *J. Polym. Sci., Polym. Phys. Ed.*, 6, pp. 1209; Moerkerke, R., Koningsveld, R., Berghmans, H., Dušek, K., Šole, K. Phase Transitions in Swollen Networks (1995) *Macromolecules*, 28 (4), pp. 1103-1107.

Of course the studies referred to in (4) and (5) did not use the fancy language, but lay the scientific basis for the materials tackled by the current article under review. In conclusion, the introduction and references to the prior art here must include proper references and a short account of the relevant and known/published physical background. The systems here must be placed in this perspective. The current submission then discusses "thermally trainable hydrogels", which are similar to gels that were described and studied by the aforementioned authors. "Positive" and "negative" "training" are perhaps interesting new ways of approaching responsive phenomena, but again, before considering publication, the basic concepts must be placed in the known literature and known physics. When reworking the manuscript, attention must be paid also to concepts such as non-ergodicity.

The specific systems described and discussed here (including the "trainable" actuators) from the materials chemistry point of view, deserve publication as an interesting addition to the exciting field of responsive hydrogels. But before further considering this submission, the manuscript must be thoroughly revised, the observations must be placed in the known context, and the mentioned concepts and prior knowledge must be properly considered.

Point-by-point responses to reviewers' comments

General response: We would like to thank all reviewers for taking their time in reviewing our manuscript and for their constructive comments and suggestions that greatly helped improve the quality of our work. We have tried our best to address their comments as listed below in our point-by-point responses. In addition, we have by ourselves identified and revised a mistake in the calculation of the degree of substitution in the Supplementary Information as shown below.

Calculation of degree of substitution. Appearance of proton signals from acryloyl groups ($-\text{CO}-\text{CH}=\text{CH}_2$) in ^1H NMR spectrum between 5.9 ppm and 6.4 ppm indicated the partial acrylation of agarose¹. The degree of substitution was calculated from the integral ratio of the signals at 5.9–6.4 ppm compared to that of the signals from protons at position 1 in 3,6 anhydro- α -L-galactopyranose unit of agarose (red highlighted in Supplementary Fig. 15) at around 5.1 ppm, which were used as internal standard².

Supplementary Table 3. Feed ratio and degree of substitution calculated from NMR spectra.

Feed ratio	10 %	15 %	30 %	50 %
Degree of substitution	0.7 %	1.1 %	2.3 %	2.8 %

Supplementary Figure 15. Reaction scheme for the synthesis of acrylated agarose and the ^1H NMR spectra of different degrees of acrylation in agarose. **a** Acrylation reaction in agarose. **b** Pure agarose and acrylated agarose with different DA (10, 15, 30, and 50).

Reviewer #1:

Recommendation: Publish in Nature Comm after minor revisions (almost as is)

The article by Hu et al. reports the design of thermally trainable hydrogel systems based on a double or mixed network design. The volumetric response of the system can either increase in volume (positive training) or decrease in volume decrease (negative training) through a training process induced by temperatures above either the cloud point of the PNIPAm network or above the melting point of the contained agarose network.

Positive training is achieved by heating the dual networks, melting the agarose, and diffusing the agarose out of the networks, which allows the PNIPAm network to swell and de-swell more freely and results in a larger swollen state.

For negative training, a mixed network containing PNIPAm and acrylated agarose is used. Due to chemical cross-linking, the agarose coils become entrapped in the hydrogel upon heating and contraction. Upon cooling, the agarose gels faster than the PNIPAm reswells, so it is trapped in a swelling-restricted state, resulting in an overall shrunken state. Using tensile tests, the authors demonstrated that mechanical properties can be modified by temperature-induced training. Here, positive training leads to softer hydrogels and negative training leads to tougher and stronger hydrogels. For demonstration, the authors used a gripper consisting of a bilayer of both trainable systems, and thus making a gripper that must first needs to be conditioned at high temperatures to be fully functional at lower temperatures. Overall, this is an excellent paper that clearly presents its content. The data provided data is nearly flawless. Although it may be bold to simply to wash agarose out of a double network and call this positive training, it is applied in a clever way. The bilayer actuator that needs to be conditioned to work at lower temperatures is impressive.

Our answer: We thank the reviewer for the positive assessment and perfect summary of our work. We have further revised the manuscript based on the reviewer's comments as listed below.

Minor points:

1. You state that you are using 0.05 mol% PEGDA in relation to NIPAm. Is this an optimized ratio? Because the CL density impacts the swelling of pNIPAm and thus the training ratio of the positive training hydrogels. If you have data on this, it should be provided in the supporting information.

Our answer: We thank the reviewer for the suggestion. Indeed the crosslinking density affects the swelling of PNIPAm and thus the training process. The reason for choosing the 0.05 mol% PEGDA was that it provides an optimized combination of mechanical properties and training effects. For lower CL density, the hydrogels became very soft after training, while further increasing the CL density did not bring much change in the training effect but slightly inferior mechanical properties before training. We have now added new data of the positive training hydrogels containing 0.025 & 0.1 mol% PEGDA relative to NIPAm in Supplementary Figure 9 (change of swelling ratios during training) and Supplementary Figure 10 (mechanical properties before and after training). We have also added the following clarifying text to the manuscript.

The concentration of PEGDA is optimized to achieve the best combination of training effect and mechanical properties (Supplementary Figs. 9&10). Lower crosslinking density results in higher training effect but mechanically rather weak hydrogels, while increasing the crosslinking density to 0.1 mol% results in similar training effect compared to 0.05 mol% (Supplementary Fig. 9), but lower mechanical strength before training (Supplementary Fig. 10).

Supplementary Figure 9. Effect of crosslinking density on the swelling ratio in positive training hydrogels. The hydrogels containing 2 wt% agarose were prepared with different crosslinking density: 0.025 mol% or 0.1 mol% PEGDA relative to NIPAm. Error bars are standard deviations from 4 measurements.

Supplementary Figure 10. Effect of crosslinking density on the mechanical properties in positive training hydrogels. **a** Tensile tests of pristine hydrogels before training. **b** Tensile tests of hydrogels after training for 3 cycles at 70°C. The hydrogels containing 2 wt% agarose were prepared with different crosslinking density: 0.025 mol%, 0.05 mol% or 0.1 mol% PEGDA relative to NIPAm (10 wt%).

2. With respect to Fig. S6: One might wonder why the initial swelling ratio at rt is not the same before heating to 40 and 70 °C.

Our answer: We thank the reviewer for the careful examination. The swelling ratios are different before heating to 40 and 70°C, because the sample heated to 70°C was pre-trained for 3 cycles at 70°C, while the sample heated to 40°C was the pristine hydrogel. We apologize for not making the difference clear in our figure caption. Besides, we have noticed minor errors in our original Supplementary Fig. 6a regarding the initial swelling ratio at RT before heating to 70°C, which was due to the normalization to a wrong initial disk diameter. We have now updated Supplementary Fig. 6, unified the style of Supplementary Fig. 4 on the kinetics of negative training hydrogels, and updated the kinetics fitting in Supplementary Table 1, as shown below.

Supplementary Figure 4. Shrinking and swelling kinetics of negative training hydrogels. a Heating from RT to 40 or 70°C. Note that the sample heated to 70°C was pre-trained for 3 cycles before the measurement. The sample heated to 40 °C was the pristine hydrogel. **b** Cooling from 40 or 70°C to RT. Note that the sample cooled from 70°C to RT was pre-trained for 3 cycles before the measurement. The sample cooled from 40 °C to RT was the pristine hydrogel. Hydrogels were prepared with 2 wt% ac-agarose (DA 15) and 10 wt% PNIPAm. Error bars represent standard deviations from 4 measurements.

Supplementary Figure 6. The swelling and shrinking kinetics of positive training hydrogels. a Heating from RT to 40 or 70°C. Note that the sample heated to 70°C was pre-trained for 3 cycles before the measurement. The sample heated to 40 °C was the pristine hydrogel. **b** Cooling from 40 or 70°C to RT. Note that the sample cooled from 70°C to RT was pre-trained

for 3 cycles before the measurement. The sample cooled from 40 °C was the pristine hydrogel. DN agarose/PNIPAm hydrogels containing 2 wt% agarose were used. Error bars represent standard deviations from 4 measurements.

Supplementary Table 1 Fitted t_1 for swelling/shrinking kinetics of hydrogels

	t_1 – shrinking (min)		t_1 – swelling (min)	
	25 → 40 °C	25 → 70 °C	40 → 25°C	70 → 25 °C
Positive training hydrogel	1.22 ± 0.08	0.44 ± 0.03	4.53 ± 1.16	11.87 ± 3.21
Negative training hydrogel	1.37 ± 1.01	0.11 ± 0.13	7.58 ± 0.05	7.29 ± 0.91

Great Paper! Congratulations

Our answer: We thank the reviewer for the great comments.

Reviewer #2:

In this manuscript, the authors demonstrate a thermally trainable hydrogel system based on Agarose/PNIPAm DN gel. Either positive or negative training can be achieved, and the mechanical properties after training lead to soften, stiffen or toughen the gels. The soft hydrogel actuators were also prepared based on the bilayer gel of positive and negative gels. The work is interesting, and the design is ingenious. I'd like to recommend to accept this work by Nature Comm after the authors addressing well the following issues:

Our answer: We thank the reviewer for the positive assessment of our manuscript. We have revised the manuscript based on the reviewer's comments as shown below.

1. Since the network of gels are mainly depended on double network hydrogels, the title using "Dual Network Hydrogels" isn't suitable.

Our answer: We thank for the suggestion. Indeed, we have given much rumination about the nomenclature of our system before the submission of the manuscript, and "Double Network Hydrogel" has been one option for the title. Eventually, we found it more appropriate to use "dual network" to represent the current system instead of the "double network". The reason is following:

1. The positive training hydrogel agarose/PNIPAm is a double network hydrogel, since the PNIPAm network is crosslinked by PEGDA, and there is no chemical bond between the two different networks, i.e., agarose and PNIPAm networks.
2. The negative training hydrogel is NOT a double network hydrogel, since the PNIPAm network is not crosslinked by PEGDA, but only by the acrylate groups on the ac-agarose. There thus exist chemical bonds between the PNIPAm and ac-agarose networks. Our naming in the original manuscript "mixed network" can also be misleading. Therefore, we have decided to rename the ac-agarose/PNIPAm network to Interconnected Interpenetrating Network (IIN) in this revision, since it is essentially an interpenetrating network hydrogel, where the two networks are interconnected by chemical bonds. See also a review in *Polymer* 2020, 207, 122929

Because the two types of hydrogels are based on different network designs and they all contained two networks, we find it more appropriate to use "dual network" in the title to represent both of our systems, instead of double network that only represents the positive training hydrogel. We have added the following clarification in the main text.

*To implement negative training, we designed an **interconnected interpenetrating network (IIN) hydrogel containing an acrylated agarose (ac-agarose) network and PNIPAm (Fig. 1h). The main difference of IIN to conventional DN or interpenetrating network hydrogel is that the two networks in the IIN are interconnected together by chemical bonds⁴⁶.***

We also apologize for the misleading illustrations in Fig. 1 e-h, which may have given the impression that the ac-agarose/PNIPAm hydrogel was also based on double network by adding ac-agarose. We have revised Fig. 1 e-h and tried to highlight the differences between the two crosslinking methods.

Fig. 1 Schematic illustrations of positive and negative training in dual network hydrogels. **a** Stimuli of varying intensities. Responses of **(b)** conventional responsive materials, **(c)** positive training material, and **(d)** negative training material induced by the stimuli in **(a)**. **e** Compositions of the trainable hydrogels. **f** A single network PNIPAm gel showing conventional thermoreversible volume changes corresponding to **(b)**. **g** A double network (DN) agarose/PNIPAm hydrogel showing positive training corresponding to **(c)**. **h** An interconnected interpenetrating network (IIN) ac-agarose/PNIPAm hydrogel showing negative training corresponding to **(d)**.

2. The training concept used in this work is much different from others because the final hydrogels were compared not with the original DN gel but with the gel after removal of agarose. If compared to the original DN gel, all the mechanical properties became very poor. Therefore, I suspect whether the concept “training” is correctly used or not?

Our answer: We believe that there might be a misunderstanding of the training process of the mechanical properties and apologize for the unclear drawings and errors in Fig. 4a-c. The final trained hydrogels were actually always compared to the original hydrogels in their pristine (as-prepared) states in these figures. We have now corrected the errors and redrawn the figure for better clarity as shown below.

Fig. 4 Training of mechanical properties. **a** Tensile tests of positive training hydrogel containing 2 wt% agarose, showing softening after training. **b** Tensile tests of negative training hydrogel containing 2 wt% of ac-agarose (DA 30), showing enhancement of Young's modulus (stiffening) after training. **c** Tensile tests of negative training hydrogel containing 2 wt% of ac-agarose (DA 50), showing increase in toughness (toughening) after training. All trainings were conducted at 70°C for 3 cycles.

For instance, the positive training DN hydrogel in Fig. 4a shows weakening of mechanical properties compared to the original DN hydrogel. In contrast, the negative training IIN hydrogels in Fig. 4 b, c show enhanced mechanical properties compared to the original as-prepared (pristine) IIN hydrogels. We thus believe that the use of training is justified due to the enhancement of the mechanical properties compared to the original hydrogels.

3. In Figure 1h, the authors introduced the ac-agarose in the hydrogel to achieve negative training. But it easily makes reader confuse. It seems that additional ac-agarose was added based on Figure 1e.

Our answer: We thank the reviewer for the suggestion and agree that the original illustration was misleading. In fact, the ac-agarose was NOT added to the DN hydrogel to make the negative training hydrogel. The negative training hydrogel was based on a distinct design using solely ac-agarose. We have thus revised the Fig. 1 e-h to make the difference more clear and highlighted the different crosslinking mechanisms in the two systems.

Fig. 1 Schematic illustrations of positive and negative training in agarose/PNIPAm dual network hydrogels. e Compositions of the trainable hydrogels. **f** A single network PNIPAm gel showing conventional thermoreversible volume changes corresponding to (b). **g** A double network (DN) agarose/PNIPAm hydrogel showing positive training corresponding to (c). **h** An interconnected interpenetrating network (IIN) ac-agarose/PNIPAm hydrogel showing negative training corresponding to (d).

4. The negative training is achieved by the rapid gelation of agarose while the slow chain dynamic of PNIPAm as the temperature > LCST. The agarose first network resists the chain extension or recovery. However, if the time is long enough, can the size of gel can be recovered to the original one?

Our answer: We thank the reviewer for the suggestions and have carried out the control experiments where the trained hydrogel was further stored at RT for more than 30 hours, which

showed no significant recovery of size compared to freshly trained hydrogel. We have now added following text and the data as Supplementary Fig. 16.

Such decrease is preserved when the temperature returns to 40°C, and the reduced equilibrium swelling ratio at 25°C remains stable for more than 30 hours (Supplementary Fig. 16).

Supplementary Figure 16. The decreased equilibrium swelling ratio after training in the negative training hydrogel remained stable for more than 30 hours. The hydrogel was prepared using 10 wt% PNIPAm and 2 wt% ac-agarose (DA 15). The training process consisted of 3 cycles of 5 min heating at 70°C and 30 min at 25°C. Error bars represent standard deviations from 4 samples.

5. In Figure 3e, the SEM before training is for ac-Agarose/PNIPAm DN gel? If so, compared to Figure 2e with Agarose/PNIPAM DN gel, why ac-Agarose/PNIPAm DN gel with chemical crosslinking show much looser network structure at the same concentration of agarose?

Our answer: Yes, the SEM image in Fig. 3e is for the ac-agarose/PNIPAm negative training hydrogel. In the ac-Agarose/PNIPAm, the PNIPAm network was not crosslinked by PEGDA, but only by the acrylate groups on the agarose. Therefore, the network structure of the ac-agarose/PNIPAm is much looser than the agarose/PNIPAm DN hydrogel that has been crosslinked by PEGDA. We apologize for the misleading Fig. 1 f-h and have now added the following clarification to the main text and revised Fig. 1 f-h accordingly (shown in the previous answer).

Compared to the agarose/PNIPAm DN hydrogel, the pores in the pristine ac-agarose/PNIPAm IIN hydrogel are larger due to the lack of PEGDA crosslinker, and the PNIPAm network in the IIN hydrogel is only crosslinked by the acrylate groups in the ac-agarose.

Reviewer #3:

(1) The abstract starts with the following sentence: “Inspired by biological systems, mechanically trainable materials have received burgeoning research interests as they harbor great potential for future adaptive and intelligent material systems.” The literature is full of cliché’s like this. Is it really necessary to start the abstract with this cliché? The reviewer has similar feelings about the use of the word “training”. The reviewer realizes that the (ab)use of renaming existing concepts is becoming an unfortunate habit in the literature. But there is no need to rename thermo-rheological (or physical) history that the material experienced to “training”. The reviewer’s strong advise is to please reconsider the language and use established, classical terminology whenever possible, instead of the fancy looking (and sounding) language that does not help understanding. This comment is just a friendly advise.

Our answer: We thank the reviewer for taking the time to review our manuscript and appreciate his/her suggestions regarding the use of classical terminology. We agree with the reviewer that there is a trend in the literature toward the misuse of language (such as smart materials) and are mindful of the abuses of concepts. We have aimed to use established, classical terminology wherever possible, such as double network hydrogels and LCST of PNIPAm, which are used throughout this manuscript. We have also changed our original naming of the ac-agarose/PNIPAm from “mixed network” to “interconnected interpenetrating network” hydrogel, to better fit with classical terminology. We have added the following clarification in the main text.

*To implement negative training, we designed an **interconnected interpenetrating network (IIN) hydrogel containing an acrylated agarose (ac-agarose) network and PNIPAm (Fig. 1h). The main difference of IIN to conventional DN or interpenetrating network hydrogel is that the two networks in the IIN are interconnected together by chemical bonds⁴⁶.***

Nonetheless, after much rumination and consideration, we believe that the term "training" is necessary to clearly and efficiently convey the new concept and research results in this particular manuscript. Our work presents a special type of response in thermoresponsive PNIPAm hydrogel, which can change its volumetric response upon phase transition depending on the composition of the hydrogel and the thermal history it has experienced. This type of response does not involve the study of conventional thermo-rheological responses, which deals with classical time-temperature superposition of the viscoelasticity of the material. In our case, we deal with the phase transition behavior that led to volumetric changes upon temperature changes below and above phase transition temperature of the hydrogel.

We acknowledge that the training can be done on thermo-rheological and mechanical properties in the other systems reported in literature. We are also aware that there is not yet a unified definition of trainable system. We have thus tried to clarify the definition of training in our introduction, taking into account the reviewer’s suggestions and the state-of-the-art systems in literature (e.g., *Science* 363, 504–508 (2019), *Proceedings of the National Academy of Sciences* 116, 10244–10249 (2019), and *Nature Materials* 20, 869–874 (2021)).

***While there does not yet exist a unified definition,** we broadly define trainable materials as materials capable of modifying their own properties and/or stimuli-responses depending on*

the intensity of stimulus previously experienced, without the participation of a different stimulus or addition of chemicals. The modified properties can be, for instance, mechanical, thermo-rheological, or optical properties, or phase transitions.

(2) However, the abstract is full of such blah-blah, and does not really serve the function of a scientific abstract. An abstract should be “..a summary of the contents of a book, article, or speech” (Oxford dictionary). This “abstract” is thus not an abstract, as there is virtually nothing about the summary of the article. Conclusion: please rewrite and make it a real abstract (what/why/how... you studied and what were the main findings/conclusions).

Our answer: We thank the reviewer for the suggestion. We have followed *Nature Communications* guideline when preparing our abstract, which reads: “*The abstract — which should be no more than 150 words long and contain no references — should serve both as a general introduction to the topic and as a brief, non-technical summary of the main results and their implications.*” We have shortened our abstract to be within the length limit and tried our best to follow the guideline in providing both “general introduction to the topic” and “brief, non-technical summary of the main results and their implications”. The detailed analysis of our new shortened abstract is shown below, with changes highlighted:

General introduction: “*Inspired by biological systems, trainable responsive materials have received burgeoning research interests for future adaptive and intelligent material systems. However, the trainable materials to date typically cannot perform active work, and the training allows only one direction of functionality change.*”

Non-technical summary of our results: “*Here, we demonstrate thermally trainable hydrogel systems consisting of two thermoresponsive polymers, where the volumetric response of the system upon phase transitions enhances or decreases through a training process above certain threshold temperature. Positive or negative training of the thermally induced deformations can be achieved, depending on the network design. Importantly, softening, stiffening, or toughening of the hydrogel can be achieved by the training process. We demonstrate trainable hydrogel actuators capable of performing increased active work or implementing an initially impossible task.*”

Implication of the results: “*The reported dual network hydrogels provide a new training strategy that can be leveraged for bio-inspired soft systems such as adaptive artificial muscles or soft robotics.*”

(3) Regarding the introduction it is very disappointing to see that the authors did not consider the pioneering work of Tanaka et al. (Annaka, M., Tanaka, T. Multiple phases of polymer gels (1992) Nature, 355 (6359), pp. 430-432, which is Tanaka’s first article in his pioneering oeuvre. The reviewer would like to point out another paper, which describes the effects of physical history (without calling “training”): Annaka, M., Tokita, M., Tanaka, T., Tanaka, S., Nakahira, T. The gel that memorizes phases (2000) Journal of Chemical Physics, 112 (1), pp. 471-477.

Our answer: We thank for the suggestions and have cited these pioneering works in corresponding places in the introduction.

Though the trainable materials should possess a memory element of previous experiences, such memory can be distinguished from other types of memories developed in responsive materials¹²⁻¹⁷, such as shape-memory¹⁸, hysteresis-based memory^{12,19}, multiple phases^{20,21}, associative memory based on self-assembly¹⁴, or swelling-induced memory¹⁵.

However, we would like to note that they are distinct from the trainable memory we demonstrate in this paper. The multiple phases are still based on hysteresis loops (though complicated ones), which does not offer permanent increase or decrease in the phase transition behavior of the gels, which is the case of our trainable hydrogels. In the introduction, we have classified the different memories in reported materials and have now added the two refs to the category of hysteresis.

The memories in these systems either require additional stimuli^{14,15} or chemicals¹⁷, or are only based on hysteresis in response to certain stimuli^{12,20,21}.

(4) Similarly disappointing that the authors did not include in their introduction and concepts, the work by Dusek et al. about the transitions in swollen polymer networks: Dusek K., Patterson D. (1968) J. Polym. Sci., Polym. Phys. Ed., 6, pp. 1209; Moerkerke, R., Koningsveld, R., Berghmans, H., Dušek, K., Šole, K. Phase Transitions in Swollen Networks (1995) Macromolecules, 28 (4), pp. 1103-1107.

Of course the studies referred to in (4) and (5) did not use the fancy language, but lay the scientific basis for the materials tackled by the current article under review. In conclusion, the introduction and references to the prior art here must include proper references and a short account of the relevant and known/published physical background. The systems here must be placed in this perspective.

Our Response: We thank for the suggestions and have cited these works in corresponding places in the main text, as highlighted below.

Typically, the responses of materials depend on the magnitude of the stimuli (Fig. 1a, b). The response is reversible and fixed, i.e., it does not depend on the history of the stimuli. Famous examples include the phase transitions in hydrogels, which have been predicted by Dusek and Patterson in 1968^{34,35} and first experimentally observed by Tanaka in 1978³⁶. The crosslinked hydrogel network is non-ergodic³⁷, and hysteresis is commonly observed during phase transitions. In particular, poly(N-isopropylacrylamide) (PNIPAm) hydrogel undergoes reversible shrinking and swelling due to its Lower Critical Solution Temperature (LCST) as shown in Fig. 1f³⁸. Here we use LCST to denote the phase transition temperatures of the PNIPAm networks in our present system.

In addition, we have tried to clarify the differences of the current system with state-of-the-art systems in our introduction, as shown below.

On the other hand, various mechanisms have been developed to post-modify or control the properties of responsive materials, such as temperature-triggered reversible hardening of a hydrogel²², photo-triggered mechanical strengthening in synergistic covalent and supramolecular polymers²³, thermally induced self-healing and strengthening in an elastomer²⁴, post-modification of sample size and conductivity in organohydrogels by feeding matrix nutrients²⁵, thermal stiffening of a hydrogel by polymer-cluster interactions²⁶,

toughening of a hydrogel by post-formation of metal-coordination complex²⁷, swelling-induced strengthening of a hydrogel by deformable nano-barriers²⁸, and toughening of a hydrogel by thermal treatment in air²⁹. Another interesting artificial muscle system capable of performing active work has shown self-strengthening upon mechanical trainings³⁰. However, the stimulus (heat) driving the deformation is different from the stimulus (deformation) triggering the mechanical strengthening, and the thermal stimulus cannot enhance the mechanical properties alone. Despite the fascinating progresses in these systems with post-modification capability, there is generally a lack of training in the material's response by the stimulus driving the response.

The current submission then discusses “thermally trainable hydrogels”, which are similar to gels that were described and studied by the aforementioned authors. “Positive” and “negative” “training” are perhaps interesting new ways of approaching responsive phenomena, but again, before considering publication, the basic concepts must be placed in the known literature and known physics. When reworking the manuscript, attention must be paid also to concepts such as non-ergodicity.

Our Response: We thank for the reviewer's suggestions. In our manuscript, we have tried our best to put the current concept in the known literature and physics, as listed below:

Regarding trainable systems: *This has inspired trainable artificial materials⁸⁻¹¹, which can adapt its mechanical properties to the experienced stimulus, in particular mechanical loads. Examples include the mechano-responsive double network (DN) hydrogels⁸, muscle-like hydrogels based on alignment of nanocrystalline domains⁹, vibration-induced strengthening of a composite organo-gel¹⁰, and cyclic stretching-induced reorganization of carbon nanotube yarns¹¹.*

Regarding systems with a memory: *Though the trainable materials should possess a memory element of previous experiences, such memory can be distinguished from other types of memories developed in responsive materials¹²⁻¹⁷, such as shape-memory¹⁸, hysteresis-based memory^{12,19}, multiple phases^{20,21}, associative memory based on self-assembly¹⁴, or swelling-induced memory¹⁵. The memories in these systems either require additional stimuli^{14,15} or chemicals¹⁷, or are only based on hysteresis in response to certain stimuli^{12,20,21}.*

Regarding systems that can post-modify their properties: *On the other hand, various mechanisms have been developed to post-modify or control the properties of responsive materials, such as temperature-triggered reversible hardening of a hydrogel²², photo-triggered mechanical strengthening in synergistic covalent and supramolecular polymers²³, thermally induced self-healing and strengthening in an elastomer²⁴, post-modification of sample size and conductivity in organohydrogels by feeding matrix nutrients²⁵, thermal stiffening of a hydrogel by polymer-cluster interactions²⁶, toughening of a hydrogel by post-formation of metal-coordination complex²⁷, swelling-induced strengthening of a hydrogel by deformable nano-barriers²⁸, and toughening of a hydrogel by thermal treatment in air²⁹. Another interesting artificial muscle system capable of performing active work has shown self-strengthening upon mechanical trainings³⁰. However, the stimulus (heat) driving the deformation is different from the stimulus (deformation) triggering the mechanical strengthening, and the thermal stimulus cannot enhance the mechanical properties alone. Despite the fascinating progresses in these*

systems with post-modification capability, there is generally a lack of training in the material's response by the stimulus driving the response.

Regarding the conventional phase transition and non-ergodicity in gels: Typically, the responses of materials depend on the magnitude of the stimuli (Fig. 1a, b). The response is reversible and fixed, i.e., it does not depend on the history of the stimuli. Famous examples include the phase transitions in hydrogels, which have been predicted by Dusek and Patterson in 1968^{34,35} and first experimentally observed by Tanaka in 1978³⁶. The crosslinked hydrogel network is non-ergodic³⁷, and hysteresis is commonly observed during phase transitions. In particular, poly(*N*-isopropylacrylamide) (PNIPAm) hydrogel undergoes reversible shrinking and swelling due to its Lower Critical Solution Temperature (LCST) as shown in Fig. 1f³⁸. Here we use LCST to denote the phase transition temperatures of the PNIPAm networks in our present system. In contrast, the trainable materials should be able to modify their response depending on the history of the stimuli, in particular when the stimulus exceeds a certain threshold.

Regarding the fundamentals of agarose: Agarose is a physically crosslinked hydrogel with thermoreversible sol-gel transitions, where nanoscopic bundles of semiflexible fibrils are formed in the gel state compared to polymer coils in the sol state^{39,40}. Typically, the sol-gel transition shows large hysteresis, where the melting temperature is dozens of degrees higher than the gelling temperature⁴¹. Agarose has been utilized to construct double network hydrogels with, e.g., polyacrylamide or PNIPAm, which showed thermoreversible mechanical properties^{42,43} or enhanced whiteness upon phase transition⁴⁴.

The specific systems described and discussed here (including the “trainable” actuators) from the materials chemistry point of view, deserve publication as an interesting addition to the exciting field of responsive hydrogels. But before further considering this submission, the manuscript must be thoroughly revised, the observations must be placed in the known context, and the mentioned concepts and prior knowledge must be properly considered.

Our Response: We thank the reviewer for finding our system interesting. We have incorporated the reviewer's comments into our new revisions as listed above, and hope that our revisions have now addressed the reviewer's concerns.

Reviewers' Comments:

Reviewer #1:

Remarks to the Author:

The authors have addressed all comments of the reviewers. I was already fully supportive of publication before.

Congratulations to the authors. A pleasure reading this article.

Andreas Walther

Reviewer #3:

Remarks to the Author:

I considered the answers given by the authors. We have a disagreement about terminology used, but this should not be a "show stopper". I accept the explanations and support publication.

Point-by-point responses to reviewers' comments

Reviewer #1

The authors have addressed all comments of the reviewers. I was already fully supportive of publication before.

Congratulations to the authors. A pleasure reading this article.

Andreas Walther

Our answer: We thank the reviewer for the support of the publication.

Reviewer #3

I considered the answers given by the authors. We have a disagreement about terminology used, but this should not be a "show stopper". I accept the explanations and support publication.

Our answer: We thank the reviewer for the support of the publication. We are happy that the reviewer accepted our explanations that we have tried our best to use established terminology wherever possible, while also justifying that our system should be denoted as "trainable" as it provides a different response compared to established systems.